

# Understanding experiments and research practices for reproducibility: an exploratory study

Sheeba Samuel and Birgitta König-Ries

[1] Heinz Nixdorf Chair for Distributed Information Systems, Friedrich Schiller University Jena, Jena, Thuringia, Germany
[2] Michael Stifel Center Jena, Jena, Thuringia, Germany

## ABSTRACT

Scientific experiments and research practices vary across disciplines. The research practices followed by scientists in each domain play an essential role in the understandability and reproducibility of results. The "Reproducibility Crisis", where researchers find difficulty in reproducing published results, is currently faced by several disciplines. To understand the underlying problem in the context of the reproducibility crisis, it is important to first know the different research practices followed in their domain and the factors that hinder reproducibility. We performed an exploratory study by conducting a survey addressed to researchers representing a range of disciplines to understand scientific experiments and research practices for reproducibility. The survey findings identify a reproducibility crisis and a strong need for sharing data, code, methods, steps, and negative and positive results. Insufficient metadata, lack of publicly available data, and incomplete information in study methods are considered to be the main reasons for poor reproducibility. The survey results also address a wide number of research questions on the reproducibility of scientific results. Based on the results of our explorative study and supported by the existing published literature, we offer general recommendations that could help the scientific community to understand, reproduce, and reuse experimental data and results in the research data lifecycle.

# INTRODUCTION

Scientific experiments are a fundamental pillar of science. The way experiments are being done has dramatically changed with the advent of devices like computers, sensors, etc., that can produce and process a tremendous amount of data. With the large input data and complex preprocessing and processing, individual experiments become so complex that often scientific publications do not (and maybe cannot) provide their full picture. As a result, it becomes difficult to reproduce the published results. Reproducibility of published results is one of the challenges faced in science in the present era (*Baker, 2016a*; *Peng, 2015*; *Hutson, 2018*; *Gundersen, Gil & Aha, 2018*; *Samuel, 2019*). According to NIST (*Taylor & Kuyatt, 1994*) and Association for Computing Machinery (*2017*), a scientific experiment is said to be *reproducible*, if the experiment can be performed to get

Corresponding author
Sheeba Samuel,
sheeba.samuel@uni-jena.de

the same or similar (close-by) results by a different team using a different experimental setup. The different conditions can be in the steps, data, settings, experimental execution environment, etc (*Samuel, 2019*). In contrast, a scientific experiment is said to be *repeatable*, if the experiment can be performed to get the same results by the same team using the same experimental setup. Different research communities have embraced different definitions of reproducibility (*Zilberman & Moore, 2020*; *National Academies of Sciences, Engineering, and Medicine, 2019*). The definition of repeatability and reproducibility introduced in *Taylor & Kuyatt (1994)*, *ACM (2017)* and *Samuel (2019)* was presented to the participants in our exploratory study and is followed throughout this paper. The Reproducibility Crisis was brought into scientific communities' attention by a survey conducted by Nature in 2016 among 1,576 researchers (*Baker, 2016a*). According to the survey, around 90% of scientists agree on the existence of a reproducibility crisis. The existence of a problem in reproducing published results in different disciplines has been confirmed by a variety of studies have been attempted in different fields to check the reproducibility of published results (*Ioannidis et al., 2009*; *Prinz, Schlange & Asadullah, 2011*; *Begley & Ellis, 2012*; *Pimentel et al., 2019*; *Raff, 2019*). To ameliorate this situation, it is imperative to understand the underlying causes.

In this paper, we conduct an exploratory study as defined by *Pinsonneault & Kraemer (1993)* to understand scientific experiments and capture the research practices of scientists related to reproducibility. The motivation for this study arises from the interviews conducted with the scientists in the Collaborative Research Center (CRC) ReceptorLight project (*Samuel et al., 2017*) as well as a workshop (*BEXIS2, 2017*). These interviews provided insights on the different scientific practices followed in their experiments and their effects on reproducibility and data management. This led us to expand our study to more participants outside of this project. The aim of this study is to explore the factors that hinder reproducibility and to provide insights into the different experiment workflows and research practices followed and the general measures taken in different disciplines to ensure reproducibility. To achieve our aim, we define the following research questions (RQs) which structure the remainder of this article:

1. What leads to a reproducibility crisis in science?
2. What are the different experiment workflows and research practices followed in various fields?
3. What are the current measures taken in different fields to ensure reproducibility of results?
4. Has the introduction of FAIR data principles (*Wilkinson et al., 2016*) influenced the research practices?
5. Which research practices could improve reproducibility in general?

We address the research questions through an online survey. After the initial filtering of 150 participants, information from 101 participants was assembled for the analysis of the results. The results from the study provide insights into the ongoing existence of a reproducibility crisis and how to tackle this problem according to scientists.

In the following sections, we provide a detailed description of our findings. We start with an overview of the current state-of-the-art ("Related Work"). We describe the methods

and materials used in our survey ("Methods"). In the "Results" section, we describe our findings related to reproducibility and research practices based on the survey responses. In the "Discussion" section, we discuss the implications of our results, the limitations of our study, and provide recommendations for conducting reproducible research. We conclude the article by highlighting our major findings in the "Conclusion" section.

## RELATED WORK

Reproducibility has always been important in science as it supports extending and building on top of others' works, thus promoting scientific progress. It also helps scientists to conduct better research, allowing them to check their own results and verify the results of others, thus increasing trust in the scientific study. However, reproducibility has been a challenge in science even in the time of Galileo (1564-1642) (*Atmanspacher & Maasen, 2016*). Concerns on the drop in the quality of research have also been raised throughout the history of science (*Fanelli, 2018*; *Shiffrin, Börner & Stigler, 2018*). The assertion that many published scientific studies cannot be reproduced after several studies attempted to reproduce them (*Ioannidis et al., 2009*; *Prinz, Schlange & Asadullah, 2011*; *Nekrutenko & Taylor, 2012*; *Begley & Ellis, 2012*; *Pimentel et al., 2019*; *Raff, 2019*), has recently led the scientific community to look into the problem more seriously. Several reports have raised reproducibility concerns in genetics (*Hunt et al., 2012*; *Surolia et al., 2010*), genomics (*DeVeale, Van Der Kooy & Babak, 2012*; *Sugden et al., 2013*), and oncology (*Begley & Ellis, 2012*). While the reproduction efforts have often focused on biology, medicine, and psychology, the recent survey by Nature (*Baker, 2016a*) has shown the problem is widespread and not just pertains to specific fields (*Henderson, 2017*). These studies show that reproducibility is lacking and has impacts on scientific progress and trust in scientific results. This points to the lack in reproducibility seriously threatening scientific progress. Usage of the term "reproducibility crisis" thus seems justified, following Merriam-Webster's definition of a crisis as "a situation that has reached a critical phase". However, there is another view that this crisis narrative is partially misguided (*Fanelli, 2018*; *Shiffrin, Börner & Stigler, 2018*; *Jamieson, 2018*). *Fanelli (2018)* portrays science as facing "new opportunities and challenges" or a "revolution". *Shiffrin, Börner & Stigler (2018)* comment that irreproducibility is an old problem and science has evolved despite the problems of reproducibility. *Jamieson (2018)* comments that 'science is broken/in crisis' narrative is an overgeneralization and recommends to increase the role of self-correction in protecting the integrity of science. Whether or not to describe the problems of reproducibility as a crisis is still questionable. However, this reproducibility problem has created new challenges and perspectives that the scientific community is striving to address for improving and promoting good science.

Scientists have provided different definitions of the term reproducibility (*Taylor & Kuyatt, 1994*; *Goodman, Fanelli & Ioannidis, 2016*; *ACM, 2017*; *Plesser, 2018*; *ACM, 2020*) and a standard definition is still not agreed upon (*Baker, 2016b*). Reproducibility and replicability are often interchangeably used by scientists. *Plesser (2018)* provides a history of the definition of confusing terms: reproducibility and replicability. The *National Academies*

*of Sciences, Engineering, and Medicine (2019)* defines reproducibility as obtaining consistent computational results using the same input data, steps, methods, code, and conditions of analysis. According to NIST (*Taylor & Kuyatt, 1994*) and the Association for Computing Machinery (*2017*), reproducibility is the capability of getting the same (or close-by) results whenever the experiment is carried out by an independent experimenter using different conditions of measurement which includes the method, location, or time of measurement. We define a scientific experiment as reproducible if the experiment can be performed to get the same or similar (close-by) results by making variations in the original experiment (*Samuel, 2019*). The variations can be done in one or more of the variables like steps, data, settings, experimental execution environment, agents, order of execution, and time. This definition is also inline with the definitions of NIST (*Taylor & Kuyatt, 1994*) and the Association for Computing Machinery (*2017*). We use and validate this definition using different approaches like ontologies (*Samuel et al., 2018*), reproducibility tools like ProvBook (*Samuel & König-Ries, 2018*). The definition of repeatability and reproducibility introduced in (*Taylor & Kuyatt, 1994*; *ACM, 2017*; *Samuel, 2019*) was presented to the participants in our exploratory study and is followed throughout this paper. However, ACM recently agreed that its definitions for reproducibility and replicability were confusing (*ACM, 2017*) and have come up with a new version (*ACM, 2020*). In their new version, they define reproducibility to be performed by different team using same experimental setup.

Many studies and surveys have been conducted in different fields to identify the existence of a reproducibility crisis and check the reproducibility of published results. The existence of the reproducibility crisis is discussed in several papers belonging to different disciplines (*Nekrutenko & Taylor, 2012*; *Baker, 2016a*; *Peng, 2015*; *Hutson, 2018*; *Gundersen, Gil & Aha, 2018*; *Samuel, 2019*). The survey by Nature in 2016 (*Baker, 2016a*) brought greater insights into the reproducibility crisis by showing that 70% of 1576 researchers have tried and failed to reproduce other scientists' experiments. In a survey conducted by Nature in 2018 (*Editorial, 2018*), 86% acknowledged it as a crisis in their field, a rate similar to that found in an earlier study (*Baker, 2016a*). The survey (*AlNoamany & Borghi, 2018*) conducted among 215 participants provides insights on reproducibility related practices focusing on the usage and sharing of research software.

Many studies have also been attempted to check the reproducibility of published results by replicating studies (*Ioannidis et al., 2009*; *Prinz, Schlange & Asadullah, 2011*; *Begley & Ellis, 2012*; *Pimentel et al., 2019*; *Raff, 2019*). A study conducted by the pharmaceutical company Bayer shows that the published results from only 14 out of 67 projects were reproducible (*Prinz, Schlange & Asadullah, 2011*). There were inconsistencies between the published results and the in-house findings of the scientists at Bayer in the other projects that were not reproducible. In the study conducted by the biotech company Amgen, only 6 of 53 studies in cancer research could be reproduced (*Begley & Ellis, 2012*).

The situation in computational science is also not different. The use of computational notebooks is considered to be one of the best practices to conduct reproducible research in computational science (*Kluyver et al., 2016*). However, a study on the reproducibility of Jupyter notebooks publicly available in Github indicates that 24.11% of the notebooks

were reproducible, and only 4.03% of them had the same results as the original run (*Pimentel et al., 2019*). The failure in reproducing notebooks is due to the exceptions that occurred during their execution. *ImportError*, *NameError*, *ModuleNotFoundError*, and *FileNotFoundError* were some of the most common exceptions that resulted in the failure in the execution of many notebooks. The reason why only 4.03% of the successfully executed notebooks had the same results as the original run is not clearly mentioned in the study. However, they point out that in their study, they executed the cells in the execution order of the users and not in the traditional top-down cell order. The execution order of cells can influence the results. Another recent attempt in reproducing 255 papers from Machine Learning Research shows that just 63.5% of the papers could be successfully replicated (*Raff, 2019*). The difficulty in reproducing results has resulted in the development of many tools to help scientists in this process (*Goecks, Nekrutenko & Taylor, 2010*; *Chirigati, Shasha & Freire, 2013*; *Liu et al., 2015*; *Boettiger, 2015*; *Piccolo & Frampton, 2016*; *Project Jupyter et al., 2018*; *Samuel & König-Ries, 2020*). ReproduceMeGit (*Samuel & König-Ries, 2020*) is one such tool which analyzes the reproducibility of any GitHub repository containing Jupyter Notebooks and provides information on the number of notebooks that were successfully reproducible, those that resulted in exceptions, those with different results from the original notebooks, etc. These studies and works clearly indicate the continued existence of a problem in reproducing published results in different disciplines.

As a result of many failed reproducibility attempts, the scientific community has suggested several guidelines and recommendations to conduct reproducible research (*Research, 2014*; *Wilkinson et al., 2016*; *Knudtson et al., 2019*; *Sandve et al., 2013*; *Samsa & Samsa, 2019*). Journals like Nature ask the authors to provide the data used for experiments mentioned in the publications as a mandatory requirement. Nature introduced a reporting checklist in 2014 requiring the authors to "*make materials, data, code, and associated protocols promptly available to readers without undue qualifications*" (*Research, 2014*). The FAIR data principles introduced in this regard provide a set of guiding principles to enable findability, accessibility, interoperability, and reuse of data (*Wilkinson et al., 2016*). The National Institute of Health (NIH) provides the "Rigor and Reproducibility" guidelines to support reproducibility in biomedical research. *Knudtson et al. (2019)* survey on the factors to perform rigorous and reproducible research. *Sandve et al. (2013)* provide ten simple rules to conduct reproducible computational research. Many approaches have been provided to ensure quality of research data for reproducibility (*Simeon-Dubach, Burt & Hall, 2012*; *Plant & Parker, 2013*; *Kraus, 2014*; *Ioannidis et al., 2014*; *Begley & Ioannidis, 2015*).

In this work, we focus on understanding the research practices of scientists focusing on scientific data management and the reproducibility of results. The survey confirms the reproducibility crisis based on the perspective of researchers similar to the results from the existing literature. Inspired by the works on guidelines and recommendations to conduct reproducible research, we provide a summary of the recommendations to conduct reproducible research based on the survey questions.
## METHODS

*Participants.* We used convenience sampling for the recruitment of participants. Participation was on a voluntary basis. 150 participants responded to the survey. Only those participants who read and agreed to the informed consent form were included in the final study. Five participants who did not agree to the informed consent were excluded from the analysis. The survey was skipped by 14 participants who neither agreed nor disagreed with the informed consent were also excluded. We removed from the analysis another 14 participants who provided consent but skipped the rest of the survey. We also excluded 16 participants who provided their consent but filled only their research context and skipped the rest of the survey. This includes 2 postdocs, 7 data managers/officers, 2 students, 2 lecturers, 1 PhD student, 1 research associate, and 1 junior research group leader. They come from computer science ($n = 3$), biology ($n = 3$), physics ($n = 1$), chemistry ($n = 1$), and others ($n = 8$). Hence, participants who did not pass the initial check ($n = 49$) from 150 participants were excluded in further analyses. Responses from 101 participants were included in this study. Table 1 shows the position held by the participants at the time of answering the survey. Out of 101 respondents, the 17 others include 6 librarians, 3 software engineers, 7 data officers, and 1 publisher. The primary area of study of the participants is spread across a variety of natural sciences (Table 2). The area of study of the 26 Others include library and information science ($n = 5$), biophysics ($n = 4$), earth science ($n = 2$), social sciences ($n = 2$), behavioural science ($n = 1$), bioinformatics ($n = 1$), ecology ($n = 1$), economics ($n = 1$), electrophysiology ($n = 1$), engineering ($n = 1$), medical imaging ($n = 1$), psychology ($n = 1$), and other ($n = 5$).

*Materials.* The questionnaire was designed and developed within the framework of the CRC ReceptorLight. The author team developed the survey using three resources: (1) interviews conducted with the scientists in the CRC ReceptorLight, (2) interviews with the scientists during the workshop on "Fostering reproducible science - What data management tools can do and should do for you" conducted in conjunction with BEXIS2 UserDevConf Conference (*BEXIS2, 2017*), and (3) existing published literature on research reproducibility (*Baker, 2016a*). The interviews provided insights on the different scientific practices followed in their experiments for data management and the different challenges faced in the context of reproducibility. The literature provided details on the different aspects of reproducibility crisis factors. The questionnaire was developed in English. A group of four researchers from computer science and biology first piloted the survey before distributing it (*Pinsonneault & Kraemer, 1993*). In this step, the participants provided feedback on the length of the questionnaire, each question's priority, the clarity of the defined questions, and technical issues on filling out the questionnaire. Based on the feedback, changes were made to the final version of the questionnaire.

The survey consisted of 26 questions grouped in 6 sections. The six sections are *(1) Informed Consent Form*, *(2) Research context of the participant*, *(3) Reproducibility*, *(4) Measures taken in different fields to ensure reproducibility of results*, *(5) Important factors to understand a scientific experiment to enable reproducibility* and *(6) Experiment Workflow/Research Practices*. Table 3 summarizes the sections and the questions.

**Table 1 The current position of the participants at the time of answering the survey.**

| Current position | Count |
|---|---|
| PhD student | 27 |
| PostDoc | 18 |
| Professor | 13 |
| Data manager | 8 |
| Research associate | 7 |
| Student | 5 |
| Junior professor | 4 |
| Lecturer | 1 |
| Technical assistant | 1 |
| Other | 17 |

**Table 2 The primary area of study of the survey participants.**

| Area of study | Count |
|---|---|
| Computer science | 19 |
| Biology(other) | 17 |
| Environmental sciences | 13 |
| Molecular biology | 6 |
| Neuroscience | 6 |
| Physics | 4 |
| Plant sciences | 3 |
| Health sciences | 3 |
| Cell biology | 2 |
| MicroBiology | 1 |
| Chemistry | 1 |
| Other | 26 |

In the first and second sections, we asked the consent and the research context of participants, respectively. We used an informed consent form which consisted of information about the study's background, purpose, procedure, voluntary participation, benefits of participation, and contact information (See Questionnaire_Survey_on_Understanding_Experiments_and_Research_Practices_for_Reproducibility file for the complete questionnaire in Zenodo (*Samuel & König-Ries, 2020a*)). The invitation email, which was distributed through mailing lists, also consisted of this information. None of the questions in the survey were mandatory, apart from the informed consent form. As participants would come from different levels of knowledge on reproducibility and scientific data management, definitions of terms like 'Reproducibility', 'Reproducibility Crisis', 'Metadata', etc. were either provided on top of the sections or external links were given to their definitions.

In the third section, we asked the participants whether they think there is a reproducibility crisis or not in their research field. We presented the participants with 3 options: *Yes*, *No* and *Other* with a free text field. We provided 'Other' option with a facility to provide their

**Table 3** Summary of survey questions.

| Category | Questions content |
|---|---|
| Informed Consent Form (Datenschutzerklärung in German) | Background, purpose, and procedure of study |
| | Informed consent |
| Research context of the participant | Current position |
| | Primary area of study |
| Reproducibility | Reproducibility crisis in your field of research |
| | Factors leading to poor reproducibility |
| Measures taken in different fields to ensure reproducibility of results | Discovery of own project data |
| | Discovery of project data for a newcomer |
| | Unable to reproduce published results of others |
| | Contacted for having problems in reproducing results |
| | Repetition of experiments to reproduce results |
| Important factors to understand a scientific experiment to enable reproducibility | Experimental data |
| | Experimental requirements |
| | Experimental settings |
| | Names and contacts of people |
| | Spatial and temporal metadata |
| | Software |
| | Steps and plans |
| | Intermediate and final results |
| | Opinion on sharing other metadata |
| Experiment Workflow/Research Practices | Kind of data primarily worked with |
| | Storage of experimental data |
| | Storage of metadata |
| | Usage of scripts |
| | Knowledge of FAIR principles |
| | Implementation of FAIR principles in research |
| | Opinion on enabling reproducibility in their field |

opinion and additional comments on reproducibility crisis. The participants who either selected 'Yes' or 'Other' to this question were directed to the next question about the factors that lead to poor reproducibility from their own experiences. We presented them with 12 multiple-choice options, including 'Other' with a free text field. We chose these 12 options based on Nature's survey (*Baker, 2016a*) and our interviews and meeting with scientists in the context of the ReceptorLight project (*Samuel, 2019*). We provided the 'Other' option in most of the questions so that they could provide their opinion which is not captured in the options provided by us.

To understand the measures taken by the participants in their research field to ensure the reproducibility of results, we asked about their data management practices in the fourth section. The first question in this section was, "How easy would it be for you to find all the experimental data related to your own project in order to reproduce the results at a later point in time (e.g., 6 months after the original experiment)?". We used 5-point scale for the answer options from *Very Easy* to *Very Difficult*. We asked specifically about

the *Input Data*, *Metadata about the methods*, *Metadata about the steps*, *Metadata about the experimental setup* and *Results*. We also asked how easy would it be for a newcomer in their team to find the data related to their projects. To further understand the problem of the reproducibility crisis, we asked whether they have ever been unable to reproduce others' published results. The next question was, "Has anybody contacted you that they have a problem in reproducing your published results?". To understand the reproducibility practices of survey participants, we asked whether they repeat their experiments to verify the results.

To find out what is important for the understandability and reproducibility of scientific experiments, we asked the participants about the factors that are important for them to understand a scientific experiment in their field of research in the fifth section. We presented them with 34 factors grouped in 8 questions (see Table 3). These 34 factors have been chosen based on the concepts provided by the ReproduceMe data model (*Samuel, 2019*). The ReproduceMe is a generic data model for the representation of general elements of scientific experiments with their provenance information for their understandability and reproducibility. The data model was designed and developed with the collaborative effort of domain and computer scientists using competency questions and extended from the existing provenance models. We identified all relevant aspects when creating this data model including experiment, data, agent, activity, plan, step, setting, instrument, and material. The survey questions were built based on these factors. We also provided an open response question to describe the factors they consider important other than these 34 factors. We used 5-point scale for the answer options from *Not Important At All* to *Absolutely Essential*. We also provided 'Not applicable' option as all the factors do not apply to every participant.

In the last section, we asked about their experiment workflow and research practices. First, we asked what kind of data they work primarily with. Next, we asked about the storage place for their experimental data files and metadata like descriptions of experiments, methods, samples used, etc. To know the importance of scripts in researchers' daily research work, we asked whether they write programs at any stage in their experimental workflow. To understand the importance and acceptance of FAIR data principles (*Wilkinson et al., 2016*), we asked questions related to their awareness and use of these principles in their daily research. In the end, we provided an open response question to participants to provide comments regarding what they think is important to enable understandability and reproducibility of scientific experiments in their research field.

The online survey was implemented using *LimeSurvey (2021)*. The raw data from LimeSurvey was downloaded in Excel format. A Jupyter Notebook written in Python was used for pre-processing, analyzing, and reproducing the results. The cells in the Jupyter Notebook consist of code for the analysis of each question. The matplotlib library was used for plotting the graphs. Pandas library was used for data preparation and analysis. We used Python version 3 in the Jupyter Notebook to analyze the results. The Jupyter notebook used for the analysis of results along with the raw data and the survey questionnaire are available on Zenodo (*Samuel & König-Ries, 2020a*). The survey results can be reproduced

using Python deployed in the cloud using Binder through the GitHub website *Samuel & König-Ries (2020b)*. All data records are licensed under a CC By 4.0 License.

*Procedure.* The survey was made available online on 24th January 2019. The survey link was distributed to the scientists in the ReceptorLight project. It was also distributed to several departments at the University of Jena, Germany through internal mailing lists. Apart from the ReceptorLight project, it was also distributed among the members of the *iDiv (2021)*, *BEXIS2 (2021)* and *AquaDiva (2021)* projects. The members of the *Michael Stifel Center Jena (2021)*, which is a center to promote interdisciplinary research for Data-driven and Simulation Science also participated in this survey. It was also advertised using Twitter through the *Fusion (2021)* group account. It was also distributed through internal and public mailing lists including *Research Data Alliance Germany (2021)* and *JISCMail (2021)*.

The online survey was paginated and the progress bar was shown on each page of the survey. On the first page, the participants were first welcomed to the survey and were provided the purpose of the study, procedure, and contact information. Participants were told that the study was designed to gain a better understanding of what is needed to achieve the reproducibility of experiments in science. We informed the participants that the questions did not ask for any identification information and kept their anonymity. After reading the welcome page, the participants continued to the next page which provides the informed consent form. We provided an informed consent form with information on the General Data Protection Regulation (GDPR) (in German: Datenschutz-Grundverordnung, DSGVO). Detailed information on the background, purpose, the use of information, and procedure were provided both in English and German. We informed the participants that all the answers of the study will be published as open data in a data repository. The participants were given two options, either to agree or disagree the informed consent form. The participants who provided their consent were redirected to the survey questions. The questions of each section were provided in a single page and their progress was shown at the top of the page. When they completed the questionnaire, they were thanked for their participation and were dismissed. While, the participants who did not agree were redirected to the last page informing them that they could not continue to the survey and were dismissed. We collected only the start and last action time on the survey page of the participants who did not agree to the consent form. We do not have a measure of the survey response rate because we are not aware of the number of participants who saw the survey and chose not to respond. The average time taken by a participant to complete the survey was around 10 min.

# RESULTS

## Reproducibility crisis and its causing factors

Of 101 participants, a total of 60 (59%) think that there is a reproducibility crisis in their field of research, while, 30 (30%) of them think that there is no reproducibility crisis (Fig. S1). 11 (11%) of them selected the *Other* option and provided their opinions. Specifically, 3 participants responded that there is partly crisis. 3 others responded that they would prefer not to say the word 'crisis' instead mentioned that room for improvement

and attention is required. The others responded with comments including 'Depends on the scientific field', 'maybe', and 'I don't know'. Tables S1 and S2 further analyses the responses on the reproducibility crisis based on their position and area of study, respectively. Based on the participants' roles, we see that 20 (74%) of the total 27 PhD students and 13 (72%) of total 18 postdocs think that there is a reproducibility crisis (Table S1). In contrast, 7 (54%) of 13 professors do not believe that there is reproducibility crisis. Analyzing the area of study, 13 (68%) of 19 participants from computer science and 17 (65%) of the total 26 participants coming from molecular biology, cell biology, microbiology or biology (other) believe in the existence of reproducibility crisis (Table S2).

Figure 1 shows that the majority of the respondents consider that there is lack of data that is publicly available for use (79%), lack of sufficient metadata regarding the experiment (75%) and lack of complete information in the Methods/Standard Operating Procedures/Protocols (73%). The other reasons based on the majority votes include lack of time to follow reproducible research practices (62%), pressure to publish (61%), lack of knowledge or training on reproducible research practices (59%), lack of the information related to the settings used in original experiment (52%), poor experimental design (37%), data privacy (e.g., data sharing with third parties) (34%), Difficulty in understanding laboratory notebook records (20%) and lack of resources like equipments/devices in workplace (17%). In addition to these, 10 participants responded with other factors in the free text field. These factors include basic misunderstandings of statistics, lack of statistical understanding, type of data that cannot be reproducible, patents, copyright, and closed access, ignorance of necessity of data management, lack of mandatory pre-registration of study protocols, not following reporting guidelines, lack of collaboration, lack of automation, intrinsic uncertainty, standardised format for article preventing sufficient details to be included, and lack of funding. The responses to all the free text input field survey questions are available in Zenodo (*Samuel & König-Ries, 2020a*) (see Processed-Data_Survey_on_Understanding_Experiments_and_Research_Practices_for_Reproducibility.csv)

## Measures taken in different fields to ensure reproducibility of results

Table 4 shows the ease of findability of experimental data by the participants at a later point in time. For the survey participants, 79% of *Results* and 70% of *Input Data* are either easy or very easy to find. But when it comes to the *Metadata about the steps* (47%) and *Metadata about the experimental setup* (47%), it gets less easy. The findability of *Metadata about the steps* (36%), *setup* (38%), and *methods* (32%) shifts to neither easy nor difficult. According to the analysis, it is seen that the steps, methods, and the setup metadata are comparatively more difficult to find than the results and input data.

However, this trend changes when asked about a newcomer in their workplace to find the same experimental data of the participants without any/limited instructions from them (Table 4). The percentage of easily finding the results and input data for a newcomer drops drastically from 79% and 71% to 49% and 43%, respectively. Only 1% of *Steps* and 4% of *Experimental Setup* are very easy to find. Among all the data, the most difficult to find is the metadata about the steps and environment setup.
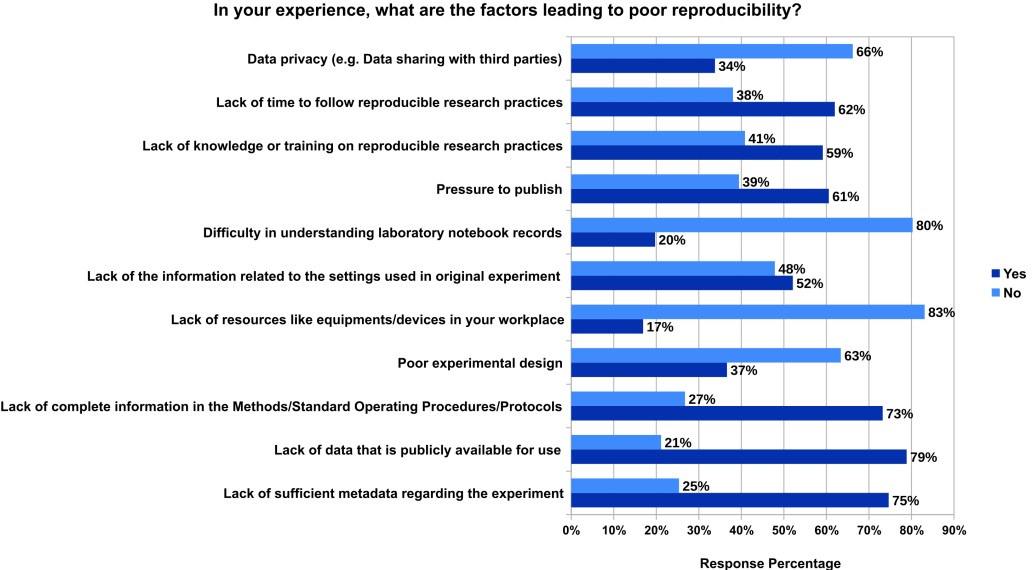

**Figure 1** The factors leading to poor reproducibility from the experience of 71 participants who fully responded to this question.

**Table 4** How easy would it be for you vs a newcomer to find all the experimental data related to your own project in order to reproduce the results at a later point in time (e.g. 6 months after the original experiment)?

|  | Findability of own data at a later point in time | | | | | Findability of own data by a newcomer | | | | |
|  | VE | E | NEND | D | VD | VE | E | NEND | D | VD |
|---|---|---|---|---|---|---|---|---|---|---|
| Input data | 29.6% | 40.7% | 18.5% | 8.6% | 2.5% | 8.3% | 34.5% | 22.6% | 23.8% | 10.7% |
| Metadata about the methods | 19.8% | 39.5% | 32.1% | 7.4% | 1.2% | 1.2% | 22.6% | 40.5% | 27.4% | 8.3% |
| Metadata about the steps | 14.8% | 32.1% | 35.8% | 13.6% | 3.7% | 1.2% | 19.0% | 32.1% | 36.9% | 10.7% |
| Metadata about the setup | 15.6% | 31.2% | 37.7% | 14.3% | 1.3% | 3.6% | 19.0% | 29.8% | 36.9% | 10.7% |
| Results | 42.0% | 37.0% | 18.5% | 1.2% | 1.2% | 8.3% | 40.5% | 27.4% | 13.1% | 10.7% |

**Notes.**
VE, Very Easy; E, Easy; NEND, Neither easy nor difficult; D, Difficult; VD, Very Difficult.

54% of them were unable to reproduce others' published results while 36% of them said 'No'. 10% of them have never tried to reproduce others' published results. Even though we see through this survey and other previous surveys (*Baker, 2016a*) that there exist issues regarding reproducibility, 95% of the participants have never been contacted, and only 5% of them have been contacted concerning issues in reproducing their published results. 53% of the respondents repeat their experiments, 12% sometimes, and 35% of them do not repeat their experiments to verify their results.

## Important factors to understand a scientific experiment to enable reproducibility

Table S3 presents the factors and the responses of the participants on the importance of sharing the factors to understand a scientific experiment to enable reproducibility. In the first question, we asked their opinion on sharing experimental data including Raw

Data, Processed Data, Negative Results, Measurements, Scripts/Code/Program, Image Annotations, and Text Annotations. Surprisingly, 80% of the participants responded that the negative results are either very important or absolutely essential while sharing data. As in the case for others, the participants consider sharing scripts (78%), processed data (73%), measurements (71%), raw data (58%), image annotations (60%), and text annotations (55%) either very important or absolutely essential.

In the next question about sharing metadata about experimental requirements, 84% of the participants consider that sharing the metadata about the experiment materials is either very important or absolutely essential. 81% of them consider the same way for the instruments used in an experiment. Regarding sharing the metadata about the settings of an experiment, participants consider that instrument settings (80%), experiment environment conditions (76%) and publications used (68%) are either very important or absolutely essential.

We asked the participants on sharing the metadata about the people/organizations who are directly or indirectly involved in an experimental study. The participants consider that it is very important or absolutely essential to share the names (70%), contacts (65%), and role (54%) of the agents who are directly involved in a scientific experiment. The participants also consider that the names (20%), contacts (18%) and role (15%) of the agents who are indirectly involved (like Manufacturer, Distributor) in a scientific experiment are very important or absolutely essential. 50% of the participants consider date as either very important or absolutely essential while 47% of them consider the same way for time. 66% of the participants consider duration as either very important or absolutely essential while 46% of them consider the same way for location. Participants consider that software parameters (80%), software version (77%), software license (37%) and scripts/code/program used (79%) are either very important or absolutely essential. Participants also consider that Laboratory Protocols (73%), Methods (93%), Activities/Steps (81%), Order of Activities/Steps (77%), Validation Methods (81%) and Quality Control Methods used (73%) are either very important or absolutely essential.

86% of the participants consider that the final results of each trial of an experiment are either very important or absolutely essential while 41% of them think the same way for intermediate results. We had asked what else should be shared when publishing experimental results for which we got 12 responses which is provided in Zenodo (*Samuel & König-Ries, 2020a*).

## Experiment workflows and research practices followed in different disciplines

The distribution of the kind of data the participants work with is shown in Fig. S2. The majority of them work with measurements (27%). The others work with images (20%), tabular data (20%), graphs (20%), and 8% of them work with multimedia files. The participants who selected the 'Other' option work with text, code, molecular, and geo-data. 30% of them store their experimental data files in the local server provided at their workplace (Fig. S3). 25% store them in their personal devices, and 21% of them specifically store in removable storage devices like hard drive, USB, etc. Only 13% of them

use version-controlled repositories like Github, GitLab, Figshare. Only 8% of them use data management platforms.

When asked about the experiment metadata storage, 58% of them use handwritten notebooks as the primary source, and 26% as a secondary source (Fig. S4). 51% of them use electronic notebooks as a primary source and 29% as a secondary source. 54% of them use data management platforms as either a primary or secondary source.

61% of the participants use scripts or programs to perform data analysis. While the other half either use them sometimes (24%) or do not use at all (15%). So in total, 85% of participants have used scripts in their experimental workflow. These participants come from not only computer science but also from different other scientific fields like neuroscience, chemistry, environmental sciences, health sciences, biology, physics, and molecular biology. The participants who do use scripts belong to environmental sciences ($n = 4$), molecular biology ($n = 3$), neuroscience ($n = 2$), biology(other) ($n = 2$), cell biology ($n = 1$), microbiology ($n = 1$), plant sciences ($n = 1$), physics ($n = 1$), and other ($n = 4$).

62% of the participants have heard about the FAIR principles, and 30% of them haven't heard about it. 8% of them have heard the term but do not know exactly what that means. It was interesting to see that the research of the participants are either always or often findable (72%), accessible (69%), interoperable (61%) and reusable (72%) (Fig. 2). We got 7 responses on what the participants think is important to enable understandability and reproducibility of scientific experiments in their field of research, which is provided in Zenodo (*Samuel & König-Ries, 2020a*).

## DISCUSSION

Reproducible research helps in improving the quality of science significantly. The existence of reproducibility crisis and the failure in reproducing published results have been brought to the attention of the scientific community through several studies in recent years (*Ioannidis et al., 2009*; *Prinz, Schlange & Asadullah, 2011*; *Begley & Ellis, 2012*; *Peng, 2015*; *Baker, 2016a*; *Hutson, 2018*; *Gundersen, Gil & Aha, 2018*; *Pimentel et al., 2019*; *Raff, 2019*). Our survey has extensively examined different aspects of reproducibility and research practices including the influence of FAIR data principles in research, the importance of factors required for sharing and reproducing scientific experiments, etc. Through our survey, we aimed to answer our research questions.

There are several key findings from our survey. The survey results show that more than half (59%) of the participants believe in the existence of a reproducibility crisis. Nature also reports that 52% of the survey participants agree that there is a significant 'crisis' of reproducibility (*Baker, 2016a*). In our survey results, there was a surprising difference in opinion between PhD students/postdocs on the one side and professors on the other with the existence of reproducibility crisis. We hypothesize that this might be due to that the PhD students and postdocs work daily with data. Though a few participants said 'crisis' is a strong word, they agreed that there is a room for improvement and considerable attention is required to support reproducibility. Pressure to publish and selective reporting were the

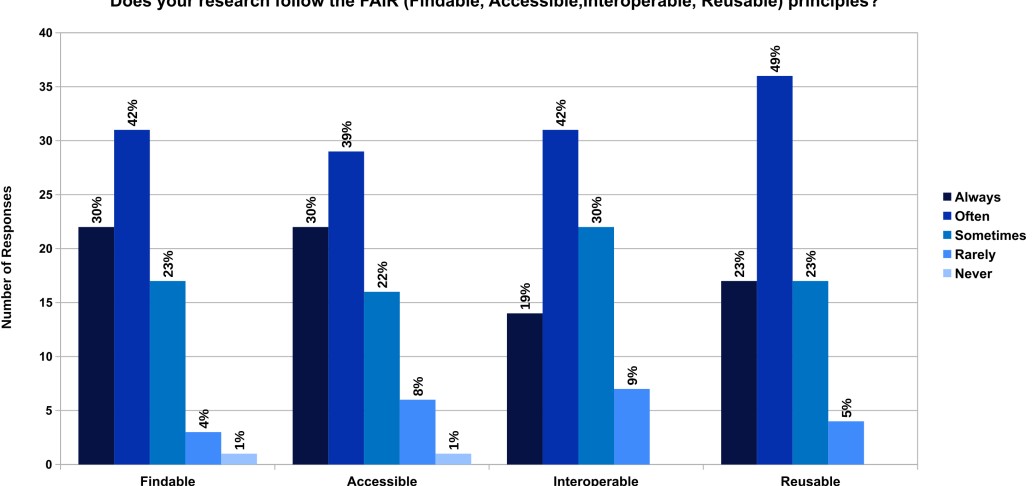

**Figure 2** Does your research follow the FAIR (Findable, Accessible, Interoperable, Reusable) principles?

primary factors that contribute to irreproducible research as reported in Nature's survey (*Baker, 2016a*). While, in our survey, lack of publicly available data, insufficient metadata, incomplete information in methods and procedures got the most mentions. This was followed by other factors like lack of time, pressure to publish, and lack of training.

Finding their own data at a later point of time is considered difficult, especially for the metadata about the methods, steps, and experimental setup. It gets more challenging in finding data for the newcomers in their workplace. The data and the steps are necessary to be documented to help both the experimenters as well as the newcomers in future. This points to the requirement of managing provenance of scientific experiments. The results present that 54% of the participants had trouble reproducing other's published results and only 5% of the respondents were contacted regarding a problem in reproducing their published results. Similar results could also be seen in Nature's survey (*Baker, 2016a*) where it was less than 20% of respondents. We assume that either people are reluctant to contact the authors or do not have the time to reproduce others' results considering the extra effort. We make such an assumption since 62% of the participants think there is a lack of time to follow reproducible research practices. We can also see that 36% of the participants have never tried to reproduce other's published results. Time is considered to be a crucial factor that affects reproducibility practices. This result is also reflected in other surveys (*Baker, 2016a*; *Harris et al., 2018*; *Nüst et al., 2018*). The other issue is the lack of training on reproducible research practices. The same number of people who think that there is a reproducibility crisis also mentioned that there is a lack of such training practices (59%). This points out the need for training of scientists on reproducible research practices. Repeatability is required to verify results, even if it is at a later point in time. 53% of the respondents repeat their own experiments to verify the results while 12% do not.

Most publications share the methods and the data that resulted in positive findings. Negative results and trials are often not mentioned in the publications as they are not

considered as accomplishments. But according to the survey, participants are keen to have the negative results being shared (*Hunter, 2017*). Participants consider experimental metadata including experimental environment conditions, instruments, and their settings, as well as experiment materials as necessary besides results and require to be shared to ensure reproducibility. 58% of the participants use handwritten laboratory notebooks as their primary source, and only 28% of them use Data management platforms as a primary source. More than half of the participants use the traditional way of documenting experimental metadata in the current era which is driven by data science. In some disciplines like biology, it is mandatory to have a handwritten lab notebook to document laboratory protocols. Even though this approach works in many disciplines, but it creates difficulty for digital preservation and reproducibility of experiments by the newcomers in the group, as pointed earlier.

Scripts are written by 85% of the participants to perform data analysis in their experimental workflow. It points out the significance of scripts in their daily research work irrespective of their scientific disciplines.

The FAIR principles introduced in 2016 are creating an impact on the research data lifecycle. 62% of the participants have heard about the FAIR principles. But 38% of them still have not heard or do not know exactly what the term means. However, more than half of the participants have tried to make their research work findable, accessible, interoperable, and reusable. Making research data interoperable by the participants was considered most challenging to follow among the FAIR principles. The survey conducted in 2018 to examine how well known or understood are FAIR principles (*ANDS, Nectar, RDS, 2018*) show similar results. In the survey, half of the respondents were already familiar with FAIR data principles and interoperability was least applied in research.

The findings from our survey show that the findability, accessibility, and reusability of data are difficult not only for their own data but also for newcomers in the team. Participants want that the metadata about the methods, steps, and experiment setup are shared in addition to the traditional sharing of results and data. It is time for the scientific community to think about the effective ways to share the end-to-end experimental workflow along with the provenance of results and implement the FAIR data principles in research.

## Limitations

There are several limitations to our study. This study was exploratory. Even though the sample is diverse for an explorative study, the findings may not be generalized to the subgroups of all the participants. Another thing that influences the survey response is the research context of participants. As part of multiple workshops and meetings conducted by the University of Jena, Germany regarding scientific data management, some of the participants from the University are aware of the concerns about reproducibility. As the survey was anonymous, we could not correlate the connection between these events and the participants. Despite these limitations, this survey provides a detailed study on scientists' views from different disciplines on the use of reproducibility practices and the important factors required for sharing metadata.

## Reproducible research recommendations

Our results show that most of the scientists are aware of the reproducibility problem. However, to fully tackle this problem, it requires a major cultural shift by the scientific community (*Peng, 2011*; *Harris et al., 2018*). Scientists can develop and promote a culture of rigor and reproducibility by following a set of best practices and recommendations for conducting reproducible research (*Brito et al., 2020*). However, this cultural shift will require time and sustained effort from the scientific community (*Peng, 2011*).

Our results report a lack of training on research reproducibility practices as one of the main factors that cause poor reproducibility. The gap in the use of research reproducibility practices might be filled by training the scientists from the beginning of their research (*Begley & Ioannidis, 2015*; *Wiljes & Cimiano, 2019*). This could be achieved by including a course on scientific data management and reproducible research practices for students and researchers in academic institutions as early as possible (*Wiljes & Cimiano, 2019*). To facilitate changes in current practices, the training should incorporate knowledge on the importance of research data management, best scientific practices for conducting reproducible research and open science, and data science practices like writing a good Data Management Plan (DMP), increase use of computational skills, etc. (*Peng, 2011*; *Fecher & Friesike, 2014*; *Michener, 2015*; *Munafò et al., 2017*; *Wiljes & Cimiano, 2019*; *Brito et al., 2020*). The training should also provide legal requirements on sharing and publishing data, copyright laws, licenses, privacy, and personal data protection (*Wiljes & Cimiano, 2019*). Our survey demonstrates that even though there is general awareness on FAIR data principles, there is a lack of awareness in implementing them in their research. In particular, how to make their research interoperable (*ANDS, Nectar, RDS, 2018*). Therefore, training should also be offered on how to implement FAIR data principles to make their data findable, accessible, interoperable, and reusable.

Another outcome shows that finding all the data is difficult not only for their own at a later point of time but also for the newcomers in their team (Table 4), and only 8% of the participants use data management platforms to store their experimental data. Without strong documentation and data management, reproducibility is challenging. The use of scientific data management platforms and data repositories help researchers to collect, manage, and store data for analysis, sharing, collaboration, and reporting (*Peng, 2011*; *Alston & Rick, 2020*). Such platforms help newcomers in the project understand and reuse the data, ensure that data are available throughout the research, make research more efficient, and increase the reproducibility of their work. However, storage medium can fail at any time, which can result in loss of data (*Hart et al., 2016*). The use of personal devices and removable storage devices to store experimental data may result in accidental failure. Therefore, it is recommended that the researchers consider and use backup services to back up data at all stages of the research process (*Hart et al., 2016*). The general public data repositories like *Figshare (2021)*, *Zenodo (2021)*, *Dryad (2021)*, *re3data (2021)*, etc., could be used by the scientists based on their scientific discipline to deposit their datasets, results, and code (*Piccolo & Frampton, 2016*). It is also favored to keep data in raw format whenever possible, which can facilitate future re-analysis and analytical reproducibility (*Sandve et al., 2013*; *Hart et al., 2016*).

The key to audit the rigor of published studies is the access to the data and metadata used to generate the results (*Brito et al., 2020*). Proper documentation of experimental workflow is one of the vital keys in successfully reproducing an experiment (*Sandve et al., 2013*). Every small detail of the experiment must be documented in order to repeat an experiment (*Ioannidis et al., 2009*; *Kaiser, 2015*). According to our survey, scientists consider sharing metadata and a clear description of raw data, negative results, measurements, settings, experimental setup, people involved, software parameters, methods, steps, and results very important to reproduce published results. It is essential that not only the positive results are published but also the negative results (*Hunter, 2017*). This is also reflected in our findings (Table S3). The provenance of results plays an important role in their reproducibility (*Missier, 2016*; *Herschel, Diestelkämper & Ben Lahmar, 2017*). The use of tools that help scientists to capture, store, query, and visualize provenance information is encouraged (*Liu et al., 2015*; *Chirigati, Shasha & Freire, 2013*; *Samuel & König-Ries, 2018*; *Murta et al., 2014*; *Boettiger, 2015*). The tools which support the reproducibility of results should be used during the documentation and publication of results. Docker (*Boettiger, 2015*), Reprozip (*Chirigati, Shasha & Freire, 2013*), Virtual machines and containers, Jupyter Notebooks (*Kluyver et al., 2016*), Binder (*Project Jupyter et al., 2018*), versioning tools are some of the examples of the tools which help in reproducing the experimental results in computational science. For the adequate documentation of experiments, the usage of general and domain-specific metadata standards for the common understanding of the data by the owners and the users are highly encouraged (*McClelland, 2003*; *Fegraus et al., 2005*; *Dublin Core Metadata Initiative (DCMI), 2012*; *McQuilton et al., 2016*). In addition to making the metadata open and discoverable, it is also recommended in FAIR data principles to use vocabularies and ontologies to ensure interoperability and reuse (*Wilkinson et al., 2016*). Several general-purpose and domain-specific vocabularies exist which aid in describing the experiments and workflows along with provenance (*Soldatova & King, 2006*; *Brinkman et al., 2010*; *Lebo et al., 2013*; *Samuel et al., 2018*).

Sharing the names, contacts, and roles of the agents involved in a scientific experiment are considered essential, as reported by our survey. The use of persistent identifiers to identify researchers (e.g., ORCID) is considered one of the good scientific practices to enable sharing information about the people, organizations, resources, and results of research (*Haak, Meadows & Brown, 2018*). Another good scientific practice is the use of permanent digital object identifiers (DOIs) for the identification of resources, including datasets, software, and results. A summary of the recommendations to conduct reproducible research through the different phases in the research data lifecycle is shown in Fig. S5.

## CONCLUSIONS

In this paper, we introduced the results of surveying scientists from different disciplines on various topics related to reproducibility and research practices. We collected the views of 101 researchers via an online survey. The analysis of the survey results confirms that the reproducibility of scientific results is an important concern in different fields of science. Lack of data that is publicly available for use, lack of sufficient metadata regarding

the experiment, and lack of complete information in the Methods/Standard Operating Procedures/Protocols are some of the primary reasons for poor reproducibility. The results show that even if the metadata about the experiments is comparatively easy to find for their own research, but the same data is difficult to be found by the newcomers or scientific community. To ensure reproducibility and understandability, it is not enough to share the input data and results, but also the negative results, metadata about the steps, experimental setup, and the methods. The results also demonstrates that even though there is general awareness on FAIR data principles, there is a lack of awareness in implementing them in their research. Based on the survey results and existing literature, we provided a set of recommendations on how to enable reproducible research.

The present study was developed to capture a broader picture of reproducible research practices. Follow up research is required to understand the different factors required in each discipline to enable reproducibility. The insights presented in this paper are based on a relatively small dataset. As the participants from this survey come from different research areas and have different roles, a more in-depth analysis of the reproducible research practices with individual roles and disciplines would reveal trends that would provide more information on tackling this problem at the grass-root level. Despite these limitations, this research offers some significant information from scientists from different disciplines on their views on reproducibility and future directions to tackle the related problems.

## ACKNOWLEDGEMENTS

We would like to thank all the participants who took part in this survey.

### Funding

This research is supported by the Deutsche Forschungsgemeinschaft (DFG) in Project Z2 of the CRC/TRR 166 High-end light microscopy elucidates membrane receptor function - ReceptorLight. The funders had no role in study design, data collection and analysis, decision to publish, or preparation of the manuscript.

### Grant Disclosures

The following grant information was disclosed by the authors:
Deutsche Forschungsgemeinschaft (DFG).

### Competing Interests

The authors declare there are no competing interests.

### Author Contributions

- Sheeba Samuel conceived and designed the experiments, performed the experiments, analyzed the data, prepared figures and/or tables, authored or reviewed drafts of the paper, and approved the final draft.

- Birgitta König-Ries conceived and designed the experiments, authored or reviewed drafts of the paper, and approved the final draft.

## Data Availability

Data is available at Zenodo:

Sheeba Samuel, & Birgitta König-Ries. (2020, May 28). fusion-jena/ReproducibilitySurvey: ReproducibilitySurvey 0.1 (Version 0.1). Zenodo. http://doi.org/10.5281/zenodo.3862597.

## Supplemental Information

Supplemental information for this article can be found online at http://dx.doi.org/10.7717/peerj.11140#supplemental-information.

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
