# Peer review of "Understanding experiments and research practices for reproducibility: an exploratory study"

_PeerJ, doi:10.7717/peerj.11140_

## Round 0.1 · original submission · Major Revisions

Please note the many occasions where reviewers agree on major revisions, and realize the full scope of the dynamics they expect within this manuscript to reach fulfillment towards further review for publication. Along with their other particular concerns, you may move forward in your decision process on eventual resubmission with their commentary in mind, and my editorial provision for opportunity to revise this manuscript accordingly, as per proper address of feedback from peer reviewers. Thank you for the opportunity to review your manuscript submission at this time, and I look forward to your prospective resubmission.

·

Basic reporting

I would like first to thank the authors to investigate a such relevant problematic, as it directly impacts reliability of so many scientific findings. The text is globally well written, although authors should take more attention of the significance, usefulness and sourcing of their assertions.
However, the study is strongly impacted by a severe gap between design, results and conclusions.
* * *
MAJOR CONCERN: UNCLEAR JUSTIFICATION AND AIM OF THE STUDY
After a very expeditious introduction, the authors present a potential target for their study that seem appropriate at line 40: “it is imperative to understand the underlying causes [of the reproducibility crisis]”. However, some methods, results and conclusion elements of a survey are presented confusingly from lines 50 to 63 to finally introduce a new goal: building a list of recommendation for good practice. At this stage, the reader is in a certain disarray. The following “Related works” paragraph gives the surprise to come back to the justification of the study. But, as the goals have already been presented, some elements are redundant with the introduction and it does not really clarify the study.

Authors should expose their study’s background and aims in an understandable way.
* * *
MINOR CONCERNS:

* Introduction / Related Works:
- We understand the use of the “reproducibility crisis” expression as resumed from other works. We would have appreciate the authors to discuss its appropriateness. As they underline themselves, the problem of reproducibility in sciences experimentation dates back to Galileo whereas the term “crisis” implies a rupture: is there one ?
- The existence of the different expressions “repeatability”, “reproducibility” and “replicability” is mentioned without lightening the reader on eventual differences, nor justifying their preference for “reproducibility”. Different approaches for reproducibility are given (l. 65-77) without discussing respective strengths and weaknesses. Authors should avoid presenting knowledge without using it for the sake of their study.
- The definition of reproducibility finally exposed is issued from authors’ prior works: is it validated ?
- Only one reported study evokes some potential cause for lack of reproducibility. We would strongly expect a further screening in this way – as it is the main subject.
- Papers reported seem mainly issued from the research area of the authors. They should enlarge their literature screening (what about qualitative approaches ?). In the same way, we expect more than two references when talking about “many tools [have been developed] l. 104.
- Many prior guidelines and recommendations are cited (l. 101 - 121): what are their strengths and weaknesses ? What is their impact ? What justifies to make another one ?

* Full paper:
- Authors should avoid describing the role of each section as the reader already is expecting it (l. 56-63, l.127-129, l.189-190, etc.).
- A better attention should be given to figures and tables title, description and correspondence to the text: table 5 do not mention the items to identify them from table 1, table 4 shows 5 columns when the authors introduce a 6-points scale at line 221, figures 5 and 8 have a title twice, etc.
- Supplementary Material is mentioned twice in the results: where can we read it ?
- A Binder repository is mentioned to reproduce the survey results, but its exact URL is not given ?
- Authors should avoid giving assertions without references (l.350, l. 403, etc.)

Experimental design
* * *
MAJOR CONCERNS:

1/ NO METHOD IS GIVEN ACCORDING TO THE RECOMMENDATION & GUIDELINES GOAL. Such a target would at least require to conduce a systematic review of literature with in-depth methods for extracting relevant and/or validated items. As a consequence, title, results and conclusions are inappropriate.

2/ RESULTS AND METHODS OF THE SURVEY ARE CONFUSINGLY MIXED, INCLUDING NUMEROUS LACKS. In particular, authors do not report the method used for the choice of the questions submitted to the participants. As a consequence, the reliability of the study cannot be assessed.
* * *
MINOR CONCERNS:

* Methods
- The “Questionnaire development” section would come up to the “Questionnaire” section.
- The questionnaire is said to include 6 sections whereas Table 1 shows 5: what about Privacy policy ?
- Table 1: the “question content” column shows summarized items whereas only a few of full items are disclosed across the Results section. Authors should deliver complete information already in the methods.
- According to the “current position”, the sum of the participants exposed in lines 174-178 only raise 91% and their label do not match Table 2. Authors should avoid duplicating data. When tables already include detailed results, text should only underline important outcomes and clarify obscure points.

* Results
- The data analysis is not exposed a priori in the methods. Line 200-208, some sub-groups percentages are given partially, without presentation of the complete data. First we may question the relevancy of percentage for such little samples. Second we wonder why such a sub-group description is not performed for other items.
- l. 205 and figure 2: we discover a questionnaire part not mentioned in the methods ! What about the “other” free text item results ?
- l. 217-223: methods in results.
- Table 5: the title mention “metadata” sharing: does this explain the 10% responses “not applicable” for raw data ? If so, what are “raw data metadata” ?
- l. 301: details ?

Validity of the findings
* * *
MAJOR CONCERNS:

1/ STRENGTHS AND WEAKNESSES ARE NOT DISCUSSED. As a consequence, the validity and the scope of the findings cannot be assessed.
- The survey link was mostly distributed within the area of the University of Jena, Germany, and only completed by a small sample of participants. To what extent these findings fit to other researchers all around the world ?
- How many excluded participants filled their research context skipping the rest of the questionnaire ? Which were their research areas ? Could it explain a recruitment bias ?
- The definition of “reproducibility” seems controversial. Why the way proposed by this study would be better than others ?
- Many guidelines have already been proposed. In what this study brings novelty ?
- The lack of time has been pointed out by participant to perform well reproducible studies. Are these recommendations in accordance with such a constraint ?
- etc.

2/ MANY RECOMMENDATIONS ARE GIVEN WITHOUT ANY JUSTIFICATION. “Storage and backup”, “use of collaboration tools”, “use of versioning systems”, etc. : even if these statements seem full of common sense, no justification for them based neither on this study’s results nor on the literature is given. As a consequence, they are unacceptable.
* * *
MINOR CONCERNS:
- we would have liked the authors discuss the particular case of sensible information when dealing with reproducibility by sharing raw data;
- many items of the given recommendations are overlapped and confusingly exposed including general principles as well as particular tools.

Additional comments

I believe this study raises very interesting and constructive findings to improve the scientific research production.
However, the results largely exceed the findings, on which authors have to fit their entire report (title, introduction, methods, results, discussion, conclusion and summary). The building of some “guidelines” in addition to the survey report and analysis is self-defeating. It causes confusions in the whole study and severely impairs quality.
Some recommendations may be given at all, provided they are based on reliable and controlled statements, and only as suggestions in the opening part of the conclusion. If authors really want to go further in this way, they should consider performing a dedicated well-conduced study.

·

Basic reporting

- Clear and unambiguous, professional English used throughout.
- Yes.
- Literature references, sufficient field background/context provided.
- Yes, although I would have expected a comparison with related work.
- Professional article structure, figures, tables. Raw data shared.
- All figures excluding Figure 9 have some minor issues, mostly resolution but sometimes a too small font.
- I also find pie charts to sometimes not being able to represent percentages fairly, but that is mostly a personal preference.
- Self-contained with relevant results to hypotheses.
- There are no hypotheses or research questions. See my comments.

Experimental design

- Original primary research within Aims and Scope of the journal.
- Yes.
- Research question well defined, relevant & meaningful. It is stated how research fills an identified knowledge gap.
- Missing. There are "investigated topics" in lines 45-49 but no RQs or hypotheses. I have several issues with this (see General comments).
- Rigorous investigation performed to a high technical & ethical standard.
- There are no information on whether an IRB/ERB was present and, if positive, whether the study was evaluated in terms of ethics.
- You declare to follow the national GDPR guidelines (DSGVO); however, one sentence worries me, and I hope that it is merely a language issue. Lines 169-171:
- "Only those participants who read and agreed to the privacy policy were included in the final dataset."
- As no question was mandatory, did you gather data in case participants did not consent to participate according to the privacy policy and discarded data afterwards? If positive, that is worrying.
- Privacy policy is IMHO a weird term that is usually related to services. As researchers, we are also limited in which "policies" we want to set, for ethical reasons. I would still call this an informed consent.
- Methods described with sufficient detail & information to replicate.
- Yes for the analysis method, although the paper misses details that are typical for survey research (response rate, number of excluded responses for each filtering criteria).
- No for the survey construction (see General comments).

Validity of the findings

- Impact and novelty not assessed. Negative/inconclusive results accepted. Meaningful replication encouraged where rationale & benefit to literature is clearly stated.
- Yes.
- All underlying data have been provided; they are robust, statistically sound, & controlled.
- The Jupyter notebook looks good to me.
- Conclusions are well stated, linked to original research question & limited to supporting results.
- They are well stated, but the study overall is not properly framed. If the study is exploratory, it should be framed as such. If the study attempts to survey a population, as wording suggests, it has many unmentioned threats to validity (see General comments).
- Speculation is welcome, but should be identified as such.
- Most the recommendations seem to come the authors personal beliefs and experiences; they do not seem to be constructed systematically.

Additional comments

The study reports on a questionnaire distributed to researchers of various fields (n=101 complete responses) towards a better understanding of the reproducibility crisis, how to tackle the crisis, and how the FAIR principles are spread.
Overall, I appreciate the initiative and several of the questions asked in the questionnaire. The results are interesting, and I was surprised to see how (relatively!) widespread is the practice to repeat own experiments with respect to my field, where this is not the case.
The paper, however, lacks several details that, in my opinion, threaten how seriously we should consider the results, which are important but need to be contextualized. The most severe concern I have is that the study presents itself to describe a population (as in descriptive study, see, e.g., Pinsonneault, A., & Kraemer, K. (1993); or norm-seeking) if not even to explain phenomena (e.g., in line 124)). That is, the paper calls readers to evaluate the study in terms of reliability, validity and, in particular for validity, representativeness and external validity. These two are weak with 101 participants, gathered from a restrict pool, to represent at least 10 roles in at least 11 areas that range from biology to physics and computer science. While the sample is diverse, which would make an exploratory study interesting, the number of people for each possible settings make comparisons meaningless to generalize. How are the recommendation even applicable with such a sample that sparkled them?
My recommendation is to declare if this was an exploratory study, which might as well be given a lack of hypotheses and research questions, and to reframe the paper as such. Research questions can, of course, be part of an exploratory study, too, and I would recommend you to add them either in place of the investigated topics (45-49) or after them.
My second major concern lies in the questionnaire itself. The paper does not offer any explanation on how it was constructed. With all existing literature on survey construction, I was expecting some words on how the items were chosen, if any framework was followed, a conceptual model, prior literature; similarly for items: how the items are linked to research questions (or goals in your case) to represent what they are supposed to represent, how the scales were chosen to represent possible answers. Sometimes you use dichotomous/binary scales with a third "other" options, sometimes ordinal scales, sometimes what seem to be Likert scales. What drove the choice? Please inform us readers better in how we can trust the numbers that come out from your questionnaire, especially in what they are supposed to represent.
In line 146, you write that "A group of four researchers from Computer Science
146 and Biology first validated the survey before distributing it." I think you meant to write "piloted" instead of "validated" here, as validation would imply particular construction and statistical tests that would enhance the way we interpret the results.
I strongly suggest to rewrite lines 166-170 and to be careful in wording. For "ethics and consent", you are reporting that you are following privacy regulations (which is good, I commend you for this), but there is little else about ethics in here. Did you gather informed consent? Was there a need for an institutional review board to approve the study? Was there one at the institute (not having one would free you from seeking approval of one)? Please elaborate on ethics and consent. I interpret lines 169-170 to mean that data was gathered regardless of agreeing to the "privacy policy" or not, and was discarded in case no agreement was provided. I do not think that this is allowed under DSGVO/GDPR. Please specify.
I appreciate how you go into details in breaking down results, offering facets, going into details. When reading the dozens of percentages that you offer between line 191 and line 310, I was excited and surprised by several of them. I even annotated how interesting it would be to to group those in lines 238-244 by discipline. Then, I remembered about the sample, so how much sense did it make in the end to know, e.g., that "72% of PostDocs think that there is a reproducibility crisis", when this means that you found almost 13 (12.96) PostDocs who believe that there is a crisis?
The Discussion section lacks a comparison with related work. It seems that you spent a considerable time reviewing related work. Was there anything that is worth mentioning in the results you obtained? I would bet there is.
I am missing how the offered recommendations (362-486) were constructed, hopefully in a systematic way. The paper should explain the link between findings and recommendations. As written, they appear to be opinions, against which one can simply argue (e.g.: 429-434, VCS is not easy, especially Git, for those not in CS-related fields. Some people might need training to use VCS.).
Minor:
I commend you for clarifying on the differences between reproducibility and repeatability, so I recommend doing the same in the abstract to help readers to better understand your contents. People are still unclear on differences behind the meaning of these terms.
Consider adding "study" before "methods" in "and incomplete information in methods" (abstract)
Please clarify the term "metadata" in the abstract.
76-77, "We have proposed a definition of reproducibility and repeatability (Samuel, 2019) which is also inline with the definitions of NIST (Taylor and Kuyatt, 1994) and the Association for Computing Machinery (Result, 2017)." On the latter point, ACM itself recently agreed that its definitions for reproducibility and replicability were confusing (see version 1.1 from https://www.acm.org/publications/policies/artifact-review-and-badging-current). Please check that you are still aligned with ACM.
100-102, "However, a study on the reproducibility of Jupyter notebooks publicly available in Github indicates that [...] only 4.03% of them had the same results as the original run (Pimentel et al., 2019)." could you report (if any) why this happened? One would expect a Jupyter notebook to run the same everywhere.
461-464, the last sentence is confusing. A DOI helps to identify resources, not to preserve them. I know several people who believe that a DOI is a magic preservation mechanism, please help me to avoid this confusion further.

References
Pinsonneault, A., & Kraemer, K. (1993). Survey research methodology in management information systems: an assessment. Journal of management information systems, 10(2), 75-105.

---

## Round 0.2 · Minor Revisions

Note APA requirements, also, for example where (Author and Author, Date) in parentheses should use an ampersand (&). Resubmission is requested based on these peer reviews noting Minor Revisions at this time. Further revisions to resubmissions may be requested prior to final acceptance, but you should be encouraged to move forward immediately to that end. Thank you.

·

Basic reporting

A few small issues here and there, please proofread the paper.

28 - things -> devices
45 - an exploratory study (Pinsonneault and Kraemer, 1993) -> as defined by Pinsonneault and Kraemer (1993)
48, 48, 265, 266, 270, and many other places -> consider putting al these links as footnotes or references. Also for 205-208
142 - once such tool -> one such tool
243 - build -> built
Table 1, Table 2 -> consider rounding percentages here as well

Disciplines are usually written lowercase (computer science, not Computer Science)

Experimental design

Thank you very much for specifying research questions. This and the conversion to exploratory study looks way better now, and reads more transparently.

I am not 100% convinced on the first part of RQ1. Given the large studies in Nature, would we really need it in a new exploratory study? As you require to answer positively to continue the survey, I propose "What leads to a reproducibility crisis in science?"

Validity of the findings

Conversion to exploratory study makes me see the report in a way better light.

Additional comments

Dear authors,

Thank you very much for following my recommendations. I am positively surprised by how much the manuscript has improved.

Apart from minor editing suggestions, I have nothing else to report.

Best regards

·

Basic reporting

The manuscript addresses an interesting and relevant topic in science. It is well written and a really nice review on what is being discussed an described with respect to reproducibility and related concepts.

While reading it, however, I had some issue with the term "crisis" that it is a bit hyperbolic. It is a term I expect in a brochure or manifest, not in a scientific paper. It is a strong word, which in the context of scientific paper requires much more substance than simple paraphrasing some citations that are included. I have seen that previous reviewers made the same remark and with I can follow the arguments put forward in the rebuttal. However, given that the rebuttal is not part of the article and the strongness of the word "Crisis" remains, I would recommend that authors to elaborate a bit more on why it is a crisis and not just an area in dire need for improvement. If the authors decide to keep the term, I would suggest elaborating a bit more on why it is a crisis.
Peng 2105 for example, argues nicely on the role of increased access to sensory data. Other citations - like this manuscript report on conducted survey's. With survey's the question remains, whether or not it is a real crisis and not a perceived crisis. e.g. as scientists do we need to change direction or better adapt to the inevitable growth and availability of technology to perform science. There are some citations (e.g. Atmanspacher and Maasen, 2016) that probably give substantial merit to using the term "crisis", however, those citations are paywalled and not part of my subscription plans and as a result, prevented me to access.

The question is whether the term "reproducibility crisis" is really needed. Without it, the paper has enough merit. It distracts from the main message. i.e. information technology brought an influx of new data and a massive scale, which needs new approaches to handle, process and disseminate. It reports back a survey on how scientists are affected with this increase of data and suggest countermeasures to improve capturing metadata to deals with this.

Experimental design

no comment

Validity of the findings

No comment

---

## Round 0.3 · accepted · Accept

Evident issues referred to publisher for final consideration.